# Long Term Outcome of Surgical Treatment of Central Introital Dyspareunia

**DOI:** 10.3390/jcm11082066

**Published:** 2022-04-07

**Authors:** Gilbert Donders, Meri Nderlita, Viktor-Jan Vertessen, Jente Reumers

**Affiliations:** 1Femicare, Clinical Research for Women, vzw Gasthuismolenstraat 31, 3300 Tienen, Belgium; iremmeri@hotmail.com (M.N.); vjvertessen@gmail.com (V.-J.V.); jente.reumers@femicare.net (J.R.); 2Department of Obstetrics and Gynecology, Regional Heilig Hart, 3000 Tienen, Belgium; 3Department of Obstetrics and Gynecology, University Hospital Antwerpen, 2650 Edegem-Antwerp, Belgium

**Keywords:** sexual dysfunction, dyspareunia, painful intercourse, penetration pain, vulvodynia, surgical treatment, sexual satisfaction, vulva vestibulitis syndrome

## Abstract

Controversies remain regarding the preferred treatment strategy for central introital dyspareunia. The primary goal of this retrospective study was to evaluate the short- and long-term outcomes after operative management of central introital dyspareunia by widening hymenoplasty. In total, 513 patients were included, with a follow-up time of 10 years. We assessed the repair of sexual activities, quality of sexual life, and the prevalence of pregnancies after the procedure. In addition, general health status was assessed using the EuroQol-5D questionnaire. Of the 513 women operated on during the period of January 2009 until December 2019, 380 (74%) agreed to participate by sending a valid response. Eighty-seven percent of the respondents mention no to moderate pain for longer than one week after the procedure, while 9.5% and 4% of patients recalled the procedure as severely or extremely painful, respectively. Seventeen percent of patients recalled a complication; 13.2% reported prolonged healing or postoperative pain, 4.7% reported infection, and 2.4% reported bleeding. Twelve months after surgery, 72% experienced no or only slight pain during penetration. We can conclude that widening plasty of the posterior hymenal rim and vestibulum, performed as an ambulatory care procedure under local anesthesia, resolves more than 70% of the central introital pain problems one year after surgery.

## 1. Introduction

Dyspareunia is defined as a complaint of persistent or recurrent pain or discomfort associated with attempted or complete vaginal penetration. In the U.S., it affects 10 to 20% of women [1]. Introital (or entry) dyspareunia is defined as pain or discomfort at the vaginal entry, whereas vaginal dyspareunia is pain or burning due to friction after painless penetration, and deep dyspareunia is pain or discomfort on deeper penetration (felt as lower abdominal pain) [2].

Introital dyspareunia can have several causes. It can be due to identifiable causes, such as a congenitally narrow hymen; congenital malformations, such as a vaginal septum; the presence of vulvovaginal infections (e.g., candidiasis, aerobic vaginitis [3], genital herpes, and surinfected genital condyloma); and abnormal healing after previous surgery or trauma. If no causative pathology can be identified, it can be due to a condition called ‘vulvodynia’ or ‘vestibulodynia’ [4,5]. Treatment is based on detecting and alleviating the specific cause, and in the case of vulvodynia, by trying different therapies.

Donders et al. distinguished a central form of introital dyspareunia (CID), which must be differentiated from the more well-known (lateral) vestibulodynia [5,6]. CID is characterized by the presence of a rim on the posterior vestibulum at 6 o’clock (hymenal remnant) that hinders normal penetration with a penis, tool, or finger. This central form of introital dyspareunia causes excruciating pain when surpassing the posterior hymenal area from the inside out and from the outside in with an examining finger and may be accompanied by a tender redness of the fossa navicularis in front of it [5]. The difference between CID and the better known provoked localized vulvodynia (PLV) is schematized in a diagram in Figure 1. While vulvodynia (also called vestibulodynia) consists of hypersensitive demarcated areas on the lateral side of the introitus vaginae, most typically at 5 and 7 o’clock and in severe cases also at 1 and 11 o’clock (so-called 4-focal disease) [7], the typical pain problem of women suffering from CID is situated over the hymenal rim at 6 o’clock (Figure 1). Often, the vestibular zone in front of it, also called the fossa navicularis, is sensitive and bruises easily when the hymenal area is challenged or stretched. Typically, the condition is associated with later sexual debut, although it is not known whether later sexual debut is the cause of the stiffness in this area or the result of its presence [6]. If conservative treatment, such as pelvic floor relaxation therapy, is not effective, non-excisional widening plasty is a valuable primary treatment option [5]. This intervention, performed under local anesthesia, prevents many women from being endlessly treated with drugs that entail serious side effects and risks, such as antifungals, antibiotics, or antidepressants [7], or from ending up needlessly operated on to take out the whole vestibulum (vestibulectomy) [8]. The latter interventions can be very useful in selected cases, but in our opinion only if the lateral vestibulodynia has been confirmed with Q-tip testing and after CID has been excluded and first properly taken care of.

Our primary endpoint was to assess the short- and long-term functional outcome and general health status after such operative management of CID. Secondly, we identified the clinical and demographic parameters that could influence the outcomes in order to improve treatment strategies and outcomes of CID even further.

## 2. Methods

### 2.1. Patients and Setting

During the 10-year period from 1 January 2009 until 31 December 2019, 853 patients underwent surgical treatment for CID in the Lower Genital Tract Disease Center of the Department of Gynecology of the Regional Hospital Tienen, Belgium. For 660 of these patients, email addresses or phone numbers could be retrieved from the Tienen administration database regarding operations performed under the term ‘vulvoplasty’ or ‘hymenoplasty’ during that period. CID was defined as pain due to the presence of a painful rim on the posterior vestibulum at 6 o’clock (hymenal remnant) when touched by a finger moved outward or inward over that area, while providing a gentle downward touch. If the patient admitted feeling pain but could still support it, the pain was scored as ‘+’, whereas if she made painful grimaces, shouted, or automatically withdrew upward on the examination table because of extreme pain, the pain was scored as ‘++’ for ‘palpation pain’. Subsequently, a second finger was introduced (or was attempted to be introduced) to test for distention pain by gently opening both fingers a few millimeters. If this was felt as painful, a score of ‘+’ was given, whereas if it was not possible to introduce a second finger, a score of ‘++’ was given for ‘distention pain’. If the sum of ‘palpation pain’ and ‘distension pain’ was 3+ or 4+, this was diagnostic for severe CID and an indication for surgical widening posterior vestibuloplasty.

All these patients were contacted by telephone or email to ask permission to send links by email concerning a commissioned electronic survey. All patients were provided with a written informed consent form for this study, which was required to be signed and sent back before receiving the questionnaire.

### 2.2. Intake Clinical Examination and Exclusion Criteria

Apart from the assessment of CID (see above), an extensive clinical examination of the vulvo-vaginal area was performed following the same systematic routine in all patients, as described below.

Lateral introital pain was assessed in all patients by gentle touch with a cotton swab on different parts of the introital area of the vulva (Q-tip test) while asking the patient to score the pain they felt from 1 to 10, 1 being touch without pain and 10 being the most severe pain imaginable. This Q-tip touch test was performed at different points of the inner labia minora (1, 3, 5, 7, 9, and 11 o’clock), and on the posterior commissure (6 o’clock). Women with moderate (scores of 4 to 6) or severe (scores of 7 or above) lateral pain (moderate/severe provoked vestibulodynia) were only excluded if they displayed normal findings upon touch or distention of the posterior commissural area (i.e., if CID was not present). Therefore, patients with combined lateral and central introital dyspareunia were still allowed to participate in the trial and were kept in the analysis.

In addition, as part of the diagnostic workup, all patients’ vaginal fluid was tested with phase-contrast wet mount microscopy to assess lactobacillary grades and the presence of bacterial vaginosis, candida, Trichomonas, atrophy, or aerobic vaginitis [9,10]. Women with any of these microscopic findings were treated first and reassessed 6 weeks later to confirm that the condition was cured and that CID was still present in order to be included.

Finally, digital palpation of the levator ani muscles was performed to assess the tension of the pelvic muscles. Women with high muscular tension but without mono-digital central palpation pain were sent to the pelvic physiotherapist for muscle relaxation exercises.

Patients younger than 16 were excluded, as well as patients who underwent vestibulectomy, in order to prevent confusion with other pathologies, such as imperforate hymen, pain upon insertion of tampons, or therapy-resistant lateral vestibulodynia without concomitant CID.

After receiving the questionnaires, an undefined number of patients noted that they could not complete the questionnaire fully as they were not or were no longer in a sexual relationship. Those who notified us of this were excluded from the analysis of the data related to sexual function and wellbeing. We may not have been aware, however, of the presence of an actual sexual partner for all respondents, as some of them may not have mentioned this explicitly. In addition, we did not inquire about lesbian relationships, although penetration pain may or may not be an issue in the sexual health of these women.

### 2.3. Intervention

The operative approach, which was always performed by the same surgeon (GD), implied a non-excisional widening plasty, typically with local anesthesia. The surgical technique used has been described elsewhere [5]. In short, the hymenal border is incised longitudinally, including the underlying muscle (Figure 2). The mucosa and skin are separated from the underlying tissues. The mucosa is then attached to the skin with separate resorbable 3.00 Vicryl sutures. The short-term results in a smaller group of patients were excellent, with 86% of patients experiencing significant improvement in their sexual function 1 to 3 years after the intervention [5].

### 2.4. Questionnaire

The questionnaires were drawn up in Dutch, English, and French and assessed demographic and clinical characteristics, quality of life, surgical experience, and outcomes of the treatment (Appendix A). The questionnaires were anonymized and identified by study numbers only, without the names, initials, national registry numbers, dates of birth, or addresses of the patients. For this reason, during the further processing of the results, no information about the clinical severity or concomitant diseases was available for analysis beyond the information patients provided in the questionnaire.

The demographic variables assessed addressed the following information: age at the time of the surgery; body mass index (BMI); smoking; the presence of debilitating diseases, such as diabetes mellitus and pulmonary disease; and cardiovascular risk factors (CVRFs). To ensure the uniformity of surgical technique, we only included women who underwent surgery performed by the same surgeon using the same surgical procedure during the whole study period.

CVRFs were categorized by assessing the following factors: age of 55 years and over, overweight (BMI more than 25), smoking, diabetes mellitus, hypertension, or current cardiovascular diseases.

We inquired about the extent of pain the patients experienced during and after the surgery with a score from 1 (not painful) to 5 (extremely painful). Subsequently, the following possible postoperative complications were solicited: operative site infection, delayed or troubled healing, persistent pain, bleeding duration longer than one week, and wound dehiscence. Patients were asked to assess sexual function and wellbeing at 3 months, 6 months, 1 year, 2 years, and more than 2 years after the intervention. The pain was evaluated with a score from 1 to 5 (1 meaning no pain and 5 meaning no penetration was possible because of severe pain).

General health issues were tested using the EuroQol-5D questionnaire (https://euroqol.org/eq-5d-instruments accessed on 28 November 2021). The date of questionnaire completion was defined as the end of follow-up. The EuroQol-5D, Dutch version, was used to assess the functional outcomes in terms of pain and disability.

Finally, we assessed whether subsequent pregnancies resulted and recorded the number of patients with vaginal deliveries after surgery.

### 2.5. Statistic Analysis

All data were statistically analyzed using the unpaired two-sample Student’s t-test for continuous variables with a normal distribution or Chi^2^ for discrete variables. Pearson’s correlation test was used for the effect of BMI, comorbidity, smoking, and anxiety on the pain score and the coitus frequency. Results with a *p*-value of less than or equal to 0.05 were considered significant.

### 2.6. Ethical Considerations

This study was performed in compliance with the principles of the Declaration of Helsinki (2008) and the principles of GCP; in accordance with all applicable regulatory requirements; and with applicable personal data protection and processing of personal data (Directive 95/46/EC and the Belgian law of 8 December 1992 on the Protection of Privacy in relation to the Processing of Personal Data). The protocol was approved by the ethics committee of RZ Tienen (Number 2020-200003-03, 3 February 2020).

## 3. Results

### 3.1. Demographic and Clinical Characteristics

We did not have recent contact information for 191 of the 853 identified patients who underwent surgery for CID. Furthermore, two patients had died, and 10 refused to participate after telephone contact. Thus, 650 patients were contacted by mail and asked to participate in this retrospective single-center cohort study. Eighty-six of these patients were excluded, as they underwent the intervention at less than a year before (short follow-up) or claimed they had another reason to be operated on other than painful intercourse (e.g., pain when bicycling, pain upon attempt to insert tampons, etc.). Furthermore, 8 patients did not meet the inclusion criteria, 10 answered that they did not remember the operation date, 26 emails bounced back because the e-mail addresses were not correct, and 7 patients refused to participate after receiving the questionnaire. Of the 513 remaining patients who were eligible to fill out the questionnaire, 380 patients (74%) sent back complete and valid responses. There was no difference in age between the responders and non-responders (*p* = 0.14). The CONSORT diagram with the identification/inclusion process is depicted in Figure 3.

The patient characteristics are shown in Table 1. The patients had a mean age of 32 years, ranging from 16 to 71 years. The mean BMI was 23.5, and a little more than half of the women were nulliparous. The median clinical follow-up time was 6 years (interquartile range (IQR): 1–10 years). Quality of life questions revealed that only a few women were less mobile, unable to care for themselves, or unable to perform regular daily activities. About one-third mentioned that they were feeling general pain or discomfort (32.1%) and one-quarter experienced feelings of depression or anxiety (26.3%).

While 87% of patients experienced no or moderate pain during the operation, in 9.5% and 4% of patients, the procedure was remembered as severely (score 4) or extremely painful (score 5), respectively (Table 2). Seventeen percent of patients recalled a complication, most of whom (13.2%) reported prolonged healing or postoperative pain, 4.7% reported infection, and 2.4% reported bleeding.

### 3.2. Long-Term Functional Outcomes

After 1 month, 41.8% of patients were nearly pain-free during penetration; 64.5% were nearly pain-free after 3 months, and this proportion further increased to 72.1% and 74.2% by 12 and 24 months post-surgery, respectively (Figure 4).

A total of 185 (49%) and 89 (23%) patients experienced no pain or slight pain or disability, respectively, during penetration 12 months after surgery (Table 3). In the subsequent years, an additional 2% of patients could have penetration without pain or with only minor pain, and 74% could have sexual activity with no or minimal discomfort 2 or more years after the surgical intervention. Of all patients, 30% had no intercourse, 57.6% had intercourse less than once a week, and 13.4% had intercourse more than once a week. Twenty-eight percent of all patients became pregnant after the surgery and delivered one or more babies.

### 3.3. Factors Affecting Outcomes

The Pearson correlations of the patient characteristics and the pain score and coitus frequency 12 months after surgery are displayed in Table 4. Smoking showed a negative correlation (*r* = −0.117, *p* = 0.02) and anxiety showed a positive correlation with pain during intercourse (*r* = 0.114, *p* = 0.03). None of the other patient characteristics were correlated with pain scores. There was also no significant correlation between coitus frequency and any patient characteristic.

## 4. Discussion

To our knowledge, we report the largest series analyzing surgical treatment for central introital dyspareunia. The main conclusion of this survey is that the cornerstone of therapy for CID, a frequent and underdiagnosed disorder, seems to be surgery to widen the posterior hymenal area. Two years after the intervention, three of four women who received operations were happy to experience little or no pain upon penetration and could enjoy sex again.

Central introital penetration pain is a common disorder that is frequently overlooked and may be misdiagnosed as provoked localized vulvodynia. In a prospective study monitoring the causes of painful introital intercourse for one year, CID comprised 28% of all causes, while lateral vestibulodynia comprised only 18% (Donders G et al., paper in preparation). In the present study, in one center, 853 patients were diagnosed with and treated for CID for 10 years, which corresponds to around two new cases being referred to our specialized clinic per week. This indicates that CID is rather frequent and should be checked for in all patients presenting with the complaint of painful intercourse.

The advantages of this study are the large number of participants and the high response rates. In addition, the use of a stable treatment technique over the full study period is an advantage, as all patients were operated on using the same surgical technique [5] performed by the same surgeon (GD). The disadvantages were the retrospective nature of the data over a period of 10 years, which inherently calls into question the completeness and correctness of the patients’ memories. Additionally, some patients may not have been able to complete the questionnaire because they did not have a sexual partner, as 30% claimed to have no sexual intercourse at all, part of which may have been due to that reason. Furthermore, some may have been in a lesbian relationship without us knowing this. In the questionnaire, we failed to inquire about those two specific items. Finally, because of the anonymization of the data, it was not possible to verify comorbidities in the medical files, which could explain some aspects of the problem of painful intercourse.

Seventy-four percent of respondents answered the invitation by sending a fully answered questionnaire. While this is a staggering result, one still can wonder why the remaining 26% failed to properly return the completed questionnaire. Among the women who did not comply, some answered they did not recall the exact date of the operation, preventing them from answering the questions. In addition, some of the women no longer remembered the details of the operation after several years, leading them to not respond. Moreover, over this long period of time, a lot of circumstances could have changed, leading to different life conditions, such as being single, fear of resuming sexual intercourse, or being in a lesbian relationship, which we were not able to control for. We knew that non-responders did not differ in age from the responders, but regarding other variables, we did not have information.

Compared to different treatment efforts for vestibulodynia [11], including vestibulecomy [12], these results are very promising, especially given the fact that the intervention is performed under local anesthesia in an ambulatory setting, and no tissue needs to be resected.

The fact that some women claimed to have only infrequent or no sex after the surgery may have many reasons. First, during the 10-year time span of the study, the drive to have sexual intercourse may have decreased, as it does the general population [13]. Furthermore, it is not known how many women no longer had a sexual heterosexual relationship and were thereby not able to answer the questions properly. As the setup of this retrospective study did not allow us to go back to the medical files, we were also unable to compare current intercourse frequencies with the frequency before the surgery. It is, however, most likely that most of the patients were refraining from sexual activity, especially with penetration, when presenting for surgery. Indeed, from studies of patients with CID in similar patient populations in our practice, we saw that the majority of patients either refrained totally from sex with penetration or had to stop the act because of severe pain in the majority of attempts [5].

For the same reason, we were not able to check for potential comorbidities causing painful intercourse. From clinical practice, as well as from our prospective survey, we learned that several causes, such as vaginal atrophy, vulvovaginal infections, increased pelvic floor tension, psychic troubles, and lateral vestibulodynia, often coexist with CID. In this study, we were not able to analyze the frequency or severity of these medical conditions, but we have to keep in mind that the 25% of women still suffering from introital pain after 2 years may have one or more such co-morbidities.

In general, the group of patients had good health, with little cardiovascular, pulmonary, and endocrinologic system malfunctioning, and almost all answered that they were self-supportive. However, from the EuroQol-5D questionnaire, it was clear that increased levels of anxiety correlated with more severe pain scores when attempting introital penetration. While increased levels of introital dyspareunia were formerly thought to be caused by automated reflexes of fear for pain and reduced sexual arousal, research has demonstrated that affective associations with sexual stimuli are experienced more negatively in women with dyspareunia than in control women [14]. Smoking, on the other hand, was negatively related to introital pain, indicating that smokers recovered more readily from the operation and had less pain during intercourse afterward. In a study addressing the general population in Colombia to screen for dyspareunia, no such relation between smoking and painful intercourse was discovered [15], but in that cross-sectional survey, no interventions were offered, and no follow-up was performed. Although wound healing could be slower and less complete because of vasoconstriction and blood vessel damage, any such effects did obviously not play a negative role in the healing process, and, on the contrary, smoking women had less pain. We hypothesize that smoking may reduce the pain sensation or that smoking women may have a more positive attitude toward postoperative pain and sexual function.

## 5. Conclusions

We discovered that widening plasty of the posterior hymenal rim, without resection of any tissue, performed in ambulatory care under local anesthesia resolved 75% of the pain problems, but one must be aware that it can take up to one or two years to achieve the maximal effect and normal sexual activity. We realize that central introital dyspareunia is a frequent problem that needs to be examined and excluded (or treated for) in each patient presenting with introital pain upon penetration attempts during sex. In future studies, this entity, along with existing co-morbidities, must be taken into consideration. For an unclear reason, smoking women recovered better, resulting in less pain during intercourse after the surgical intervention, while anxious women suffered from more pain. This should be taken into account when proposing surgical treatment.

## Figures and Tables

**Figure 1 jcm-11-02066-f001:**
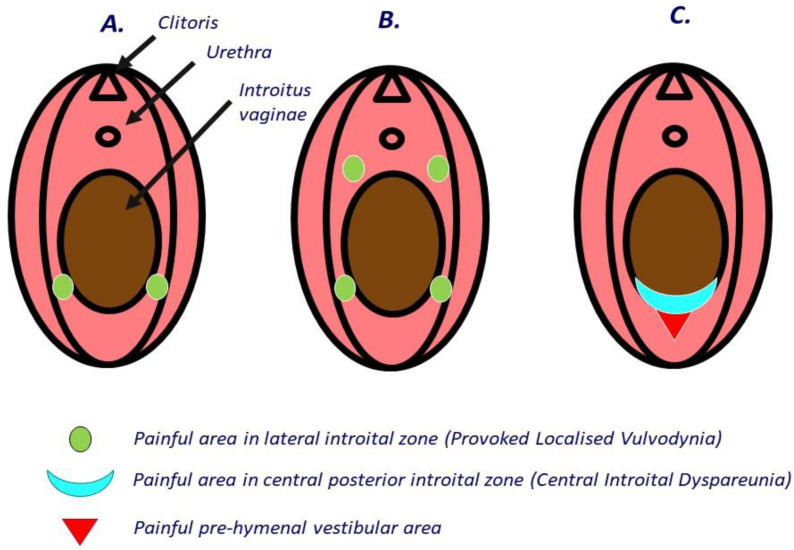
Diagram showing different types of introital pain upon penetration. (**A**) Bifocal provoked localized vulvodynia with highly sensitive/painful points (Q tip score ≥ 7/10) at 5 and/or 7 o’clock. (**B**) Four-focal disease, with extra highly sensitive/painful points at 1 and 11 o’clock adjacent to the urethral opening [7]. (**C**) Central introital pain (CID) caused by narrow, scarred, and/or bruised tissue at the posterior hymenal rim, sometimes accompanied by a painful and vulnerable fossa navicularis in front of it [5].

**Figure 2 jcm-11-02066-f002:**
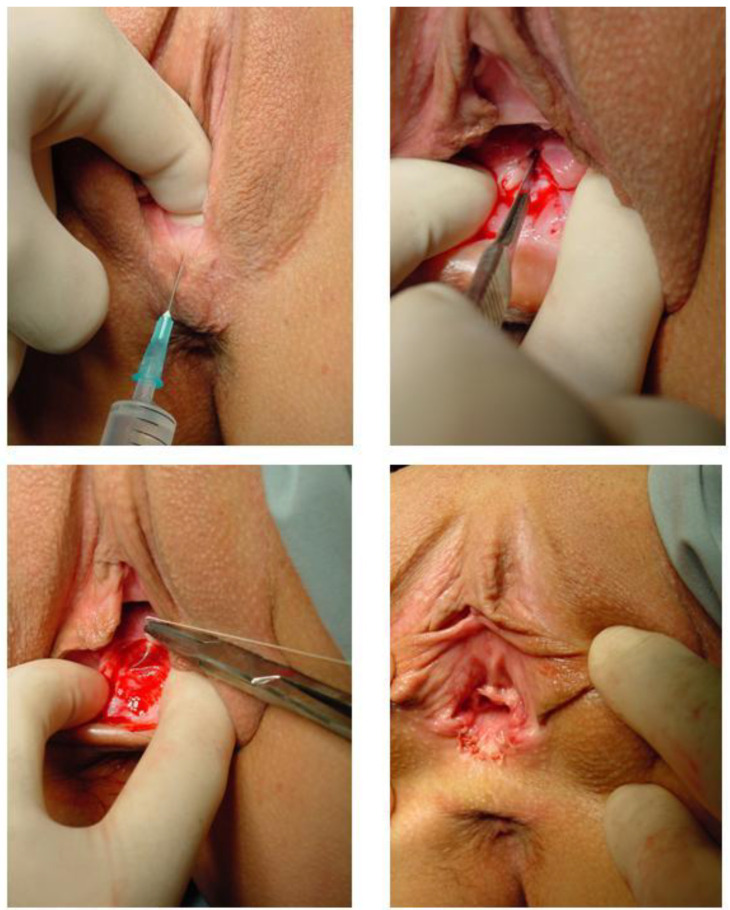
Surgical technique as first described in 2012. Additionally, after incision, the underlying tissue can be separated from the underlying tissues before suturing in order to promote easy approximation of the wound and better widening efficacy. Courtesy Donders et al., Gynaecological Surgery [5].

**Figure 3 jcm-11-02066-f003:**
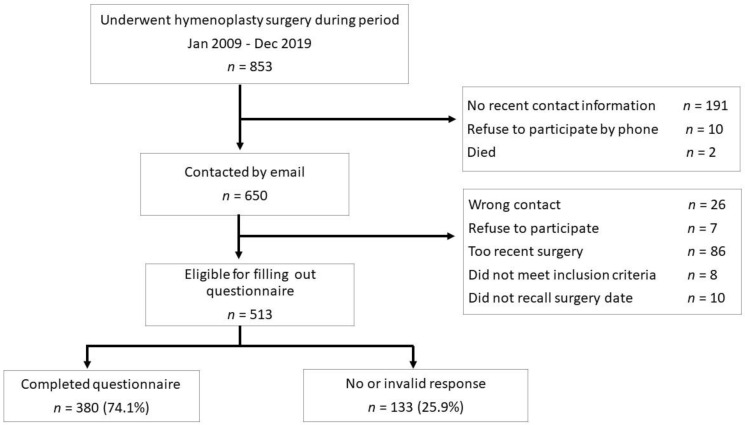
Consort diagram of the number of patients eligible for CID questionnaire after posterior vestibulum surgery (hymenoplasty) performed during the period between 1 January 2009 and 31 December 2019 in Regional Hospital Tienen, Belgium, by Prof. Dr. G. Donders.

**Figure 4 jcm-11-02066-f004:**
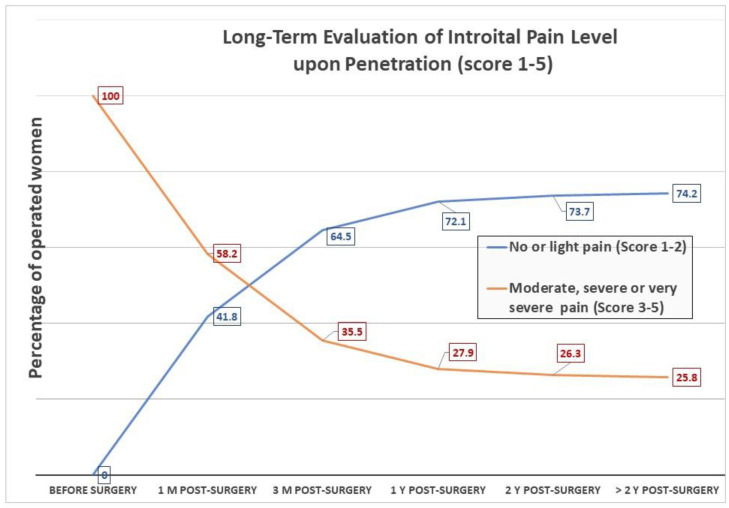
Evolution of satisfaction and pain scores upon penetration during sexual intercourse over 10 years after widening plasty of the posterior hymenal area in women suffering from central introital dyspareunia (CID). After 2 years 74.2% of patients were nearly or totally pain-free during intercourse.

**Table 1 jcm-11-02066-t001:** Questionnaire results of 531 patients operated on for central introital dyspareunia. BMI: body mass index; CVRF: cardiovascular risk factors.

Demographics	
**Age**, y, mean (median, range)	32 (27, 16–71)
**BMI**, mean (median, range)	23.5 (22.8, 16.5–49.4)
**Smoking**	150 (39.5)
**Diabetes Mellitus**	5 (1.3)
**CVRF**	19 (5)
**Parity** *p* = 0 *p* ≥ 1	199 (52.4) 181 (47.6)
**EuroQol-5D questionnaire**	
*Mobility*	
No problems	364 (95.8)
Slight problems	16 (4.2)
*Self-care*	
No problems	380 (100)
*Usual activities*	
No problems	370 (97.4)
Slight problems	10 (2.6)
*Pain/discomfort*	
No pain	258 (67.9)
Moderate pain or discomfort	112 (29.5)
Extreme pain or discomfort	10 (2.6)
*Anxiety/Depression*	
No anxiety/depression	280 (73.7)
Moderately anxious/depressed	96 (25.3)
Extremely anxious/depressed	4 (1)

**Table 2 jcm-11-02066-t002:** Assessment of pain experienced during operation and post-operative complication rate.

Pain During Surgery	
**1.** **No pain**	122 (32.1%)
**2.** **Some pain**	115 (30.3%)
**3.** **Moderate pain**	92 (24.2%)
**4.** **Severe pain**	36 (9.5%)
**5.** **Extreme pain**	15 (3.9%)
Postoperative complication rate	
No complication	314 (82.6%)
Operative site infection	66 (17.4%)
Prolonged healing	39 (10.3%)
Prolonged pain	11 (2.9%)
Bleeding with or without wound dehiscence	9 (2.4%)

**Table 3 jcm-11-02066-t003:** Experience of pain during intercourse, frequency of intercourse, and occurrence of pregnancy at short- and long-term periods after hymenal widening plasty.

Pain Score	
**One month after surgery**	
Pain score 1–2	159 (41.8)
Pain score 3–5	221 (58.2)
**6 Months**	
Pain score 1–2	245 (64.5)
Pain score 3–5	135 (35.5)
**12 Months**	
Pain score 1–2	274 (72.1)
Pain score 3–5	106 (27.9)
**24 Months**	
Pain score 1–2	280 (73.7)
Pain score 3–5	100 (26.3)
**More than 24 months**	
Pain score 1–2	282 (74.2)
Pain score 3–5	98 (25.8)
**Sexual intercourse frequency**	
<1 time a month	84 (22.1)
1–3 times a month	135 (5.5)
3–4 times a week	151 (39.8)
>4 times a week	10 (2.6)
**Pregnancy after surgery**	109 (28.6)
Vaginal delivery	93 (85.3)

**Table 4 jcm-11-02066-t004:** Multivariate analysis of risk factors predicting penetration pain and coitus frequency after hymenoplasty. Significant values are shown in bold.

		BMI	Comorbidity	Smoking	Anxiety
Pain	R^2^	0.023	0.068	**−0.117**	**0.114**
	*P*	0.662	0.185	**0.022**	**0.027**
Coitus frequency	R^2^	−0.042	0.056	0.074	−0.027
	*P*	0.418	0.277	0.15	0.600

## Data Availability

Data are available upon written request.

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
