# Peer review of "Long Term Outcome of Surgical Treatment of Central Introital Dyspareunia"

_jcm, 2022, doi:10.3390/jcm11082066_

Round 1

Reviewer 1 Report

The authors suggest that CID is a common disorder that is frequently overlooked and may be misdiagnosed as provoked vestibulodynia (or may overlap, as it is often associated with vestibular tenderness, therefore considered by others as provoked localized vulvodynia). However, this entity is not defined elsewhere, and therefore there is no data in the literature (including this study) regarding its actual prevalence.

In this light, I think that more data is required in order to better define this disorder.  The authors chose to present the data from the questionnaires, but I miss the characteristics of the population as well as more details on this diagnosis (that you probably have, due to the assessment of patients before the surgical procedure). you write that CID is "defined as pain due to the presence of a painful rim on the posterior vestibulum at 6 o'clock (hymenal remnant) when touched by a finger moved outward or inward over that area". This description also applies for women with provoked vestibulodynia which report pain when the posterior vestibule is being mechanically provoked. How do you differentiate PVD from CID?  what are the results of Q tip test (-page 2, lines 79-88)?. You write that you included patients with "lateral vestibulodynia"? were those with lateral vestibulodynia prone to surgical failure? did the surgery affect the lateral pain?

Does the problem result from an intact hymen? In this case, it may be categorized as dyspareunia in women who did not have a vaginal birth. You mention that a large part of your cohort are parous women. Does it also occur in those who gave birth? In this case, can it result from a scar tissue? Does it cause primary dyspareunia or was the pain acquired (and if secondary, due to which factors?)

Another interesting question is the cause for pain. Movement of this hymenal remanent due to friction, or another explanation?

In addition, the age range is very wide, 16-71. Were the characteristics in all women similar? In menopausal women, was it secondary to atrophy or a life-long issue?

Page 1-line 33- "congenitally narrow or imperforate hymen"- with an imperforate hymen, the presentation is hematocolpus and absence of menstruation, requiring surgery at adolescence. How can the presentation be dyspareunia?

39-40- a reference is missing (probably [5])

Page 2-66-68- the inclusion criteria or examination criteria needs clarification: was the pain produced with a gentle touch? Was it provoked only by the movement of the examining finger? What were the findings regarding vestibular tenderness?

Page 3-109-113 the surgical technique is not completely clear. Was the longitudinal incision in the 6-oclock location? can you add a figure explaining it? how deep did you incise the muscle (perineal body?)?

Table 1-

Is " Supplementary Table" (page 3, line 120) is Table 1 (methods, page 3) or Table I (results)?

This table presents results (in the Method section). Maybe move to Results.

Why are CVRF relevant?

In those who delivered, how do you explain that vaginal birth did not help this pain? Can their pain result from a scar tissue?

Page 5-178- figure I- could not find it

Legend to table II

Table II. (In) 5% and 4% (of the) patients the procedure was remembered as severely (score 4) or extremely painful (score 5), respectively. Seventeen percent of patients recalled a complication, most of which (had/reported) prolonged healing or postoperative pain (13.2%), 4.7% infection, and 2.4% bleeding.

Discussion

Page 6-209- "The main conclusion of this survey is that CID is not only frequent, underdiagnosed disorder"- you did not assess its frequency in this study, so you can't conclude that it is a frequent, undiagnosed disorder.

288- "We realize that central introital dyspareunia is a frequent problem that needs to be examined and excluded"- again, there is no published data that suggest how frequent is this disorder. In addition, defined diagnostic characteristics that the reader can use are missing.

Author Response

English language and style

( ) Extensive editing of English language and style required
( ) Moderate English changes required
(x) English language and style are fine/minor spell check required
( ) I don't feel qualified to judge about the English language and style

Yes

Can be improved

Must be improved

Not applicable

Does the introduction provide sufficient background and include all relevant references?

( )

(x)

( )

( )

Is the research design appropriate?

( )

( )

( )

( )

Are the methods adequately described?

( )

( )

( )

( )

Are the results clearly presented?

( )

( )

( )

( )

Are the conclusions supported by the results?

( )

( )

( )

( )

Comments and Suggestions for Authors

The authors suggest that CID is a common disorder that is frequently overlooked and may be misdiagnosed as provoked vestibulodynia (or may overlap, as it is often associated with vestibular tenderness, therefore considered by others as provoked localized vulvodynia). However, this entity is not defined elsewhere, and therefore there is no data in the literature (including this study) regarding its actual prevalence.

We agree with this reviewer that not enough attention has been given to CID. We want to present the present large series to show healthcare providers that trying to treat all patients with pain upon sexual penetration or attempt of penetration as PLV patients is not helpful. This condition was already defined and discussed in our former papers   (ref 5,6). Furthermore, the surgical treatment for CID was already described and assessed in a former paper following 140 patients over one year (ref 5). Unfortunately this paper was published in a non pubmed listed journal (Gynaecological Surgery), which makes it is found less often and its definition became less spread. As a result in the recent ISSVD definitions of vulvar pain, no mention is made of this condition. We hope with the present open access approach, showing a large series, followed over 10 years, we can than make up for this.

In this light, I think that more data is required in order to better define this disorder.  The authors chose to present the data from the questionnaires, but I miss the characteristics of the population as well as more details on this diagnosis (that you probably have, due to the assessment of patients before the surgical procedure). you write that CID is "defined as pain due to the presence of a painful rim on the posterior vestibulum at 6 o'clock (hymenal remnant) when touched by a finger moved outward or inward over that area". This description also applies for women with provoked vestibulodynia which report pain when the posterior vestibule is being mechanically provoked. How do you differentiate PVD from CID? 

We thank you for that analysis. It is indeed, as you correctly state, that a difference should be made between the pain over the central hymenal rim, including the triangular vestibular area just in front of it, and the lateral pain points on 5 and 7 (bifocal) or 5,7,1,and 11 o clock, which is so typical for provoked localized vulvodynia. It is also correctly stated by you that the PLV treatment is not efficient for the hymenal rim area on 5 o clock, which has a rather mechanical pathogenesis, and requires a mechanical (operative) solution. As we could demonstrate that treatment of CID is extremely successful, we are convinced that also the success of PLV treatment would dramatically improve if the CID cases would be taken out, and treated with widening posterior vestibuloplasty as we present.  To clarify this further, we included a figure to show the differences between the different types of introital dyspareunia.

what are the results of Q tip test (-page 2, lines 79-88)?.

Due to the anonymous questionnaire and methodology used in this study, we were not able to go back, to the medical files to compare the Q tip scores at 6 o clock and at 5, 7, 1 and 11 o clock. However, two different (one cross sectional and one prospective) studies were performed at our vulvovaginal disease clinic to asses these pain points separately. These studies are currently been analyzed statistically and will hopefully be published anywhere soon.

You write that you included patients with "lateral vestibulodynia"? were those with lateral vestibulodynia prone to surgical failure? did the surgery affect the lateral pain?

Indeed, again you made a very correct analysis and statement, as some patients have both PLV and CID together. In these patients CID was treated for by widening surgery of the posterior commissura, without treating the lateral PLV yet? This could have explained in part why not all patients benefited fully from the surgical treatment.  For the same reason as mentioned above, the study methodology did not permit us, however, to differentiate between combine or single pathologies. Again, the currently running prospective study will hopefully shed more light on this issue.

Does the problem result from an  and intact hymen? In this case, it may be categorized as dyspareunia in women who did not have a vaginal birth. You mention that a large part of your cohort are parous women. Does it also occur in those who gave birth? In this case, can it result from a scar tissue? Does it cause primary dyspareunia or was the pain acquired (and if secondary, due to which factors?)

Your point of criticism is well taken, and again shows your good insights in the pathology. We see CID in both nulliparous and parous women. Most of the women with primary CID (since first sexual encounter or even before, upon attempt of using tampons, are indeed nulliparous women. However also women who gave birth can have CID, which can be both primary or secondary. These women may have had a trauma with subsequent scar tissue subsequent to vaginal birth or ‘to enthousiastic’ suturing of tears or episiotomy, but more frequently also edema, lacerations and pain due to recurrent vulvovaginal infections may be the trigger to the pathogenesis of CID.

Another interesting question is the cause for pain. Movement of this hymenal remanent due to friction, or another explanation?

Again we agree that this type of pathogenesis could paly an important role. However, as this paper was intented to primarily describe treatment outcomes of the widening plasty, and the methodology did not permit us to study individual patient files, we will have to address these and other questions in more details in future studies.

In addition, the age range is very wide, 16-71. Were the characteristics in all women similar? In menopausal women, was it secondary to atrophy or a life-long issue?

Again we agree with this remark. During our systematic examination schedule of patients with painful intercourse, we ALWAYS include full analysis of the vaginal mucosal quality by fresh wet mount phase microscopy. In case of vaginal atrophy (>5% parabasal epithelial cells) this is corrected by local estrogens before any other therapy, including hymenoplasty, is installed. So, in this series, only women with CID after correction of any vulvovaginal atrophy were included.   This is outlined in the methods section of the text (L 96-102).

Page 1-line 33- "congenitally narrow or imperforate hymen"- with an imperforate hymen, the presentation is hematocolpus and absence of menstruation, requiring surgery at adolescence. How can the presentation be dyspareunia?

We corrected this and deleted ‘imperforate’.

39-40- a reference is missing (probably [5])

Ref 5 was added back. Thank you for the remark.

Page 2-66-68- the inclusion criteria or examination criteria needs clarification: was the pain produced with a gentle touch? Was it provoked only by the movement of the examining finger? What were the findings regarding vestibular tenderness?

The following was added (in yellow): ‘….while providing a gentle downward touch. When the patient admits to feel pain, but can still support it, pain is scored as ‘+’, while when she is making painful grimasses, shouts, or automatically withdraws upward on the examination table due to extreme pain, this is scored as ‘++’ for ‘palpation pain’.   Subsequently, a second finger  is intro-duced (or attempted to) to test for distention pain, by gently opening both fingers a few millimeters. If this is felt as painful a score of ‘+’ is given, while if this is not possible to introduce a second finger, it is score as ‘++’ for ‘distention pain’. If the sum of ‘palpa-tion pain and ‘distention pain’ are 3+ or 4+, this is diagnostic for severe CID and an indication for surgical widening posterior vestibuloplasty.’

Page 3-109-113 the surgical technique is not completely clear. Was the longitudinal incision in the 6-oclock location? can you add a figure explaining it? how deep did you incise the muscle (perineal body?)?

We would like to publish the figure of paper in ref 5, provided we get permission form the publisher ( was asked for).

Table 1-

Is " Supplementary Table" (page 3, line 120) is Table 1 (methods, page 3) or Table I (results)?

‘Supplement 1’ added to the text and rest eliminated

This table presents results (in the Method section). Maybe move to Results.

Table 1 was moved to results section, where it belongs.

Why are CVRF relevant?

We wanted to test if co-morbidity was another factor influcencing coital frequency in a different way than coital pain, as shown in Table 4. 

In those who delivered, how do you explain that vaginal birth did not help this pain? Can their pain result from a scar tissue?

As discussed above, some other causes can explain secondary CID, such as infections, or yet unknown causes.

Page 5-178- figure I- could not find it.

Figure 1 showing the consort diagram was added.

Legend to table II

Table II. (In) 5% and 4% (of the) patients the procedure was remembered as severely (score 4) or extremely painful (score 5), respectively. Seventeen percent of patients recalled a complication, most of which (had/reported) prolonged healing or postoperative pain (13.2%), 4.7% infection, and 2.4% bleeding.

Corrections were added. Thank you

Discussion

Page 6-209- "The main conclusion of this survey is that CID is not only frequent, underdiagnosed disorder"- you did not assess its frequency in this study, so you can't conclude that it is a frequent, undiagnosed disorder.

This is correct. However, treating 850 patients in 10 years in one single centre, means more than 2 per week, indirectly means that this condition must be frequent. We changed the phrase as follows:  The main conclusion of this survey is that the main cornerstone of therapy of CID, a frequent and underdiagnosed disorder, seems to be surgery to widen the posterior hymenal area.

288- "We realize that central introital dyspareunia is a frequent problem that needs to be examined and excluded"- again, there is no published data that suggest how frequent is this disorder. In addition, defined diagnostic characteristics that the reader can use are missing.

I agree but in this phrase we did not make allusion to the data directly showing how frequent the condition is. So I hope you agree we can keep this sentence. Diagnostic characteristics have been added (cfr above).

I had a great feeling with this review as it showed with how much expertise and passion you handle this problem, and at the same time know the literature but also feel what patients really need. Your remarks were really excellent and I cannot thank you enough to help us improve the quality of this paper dramatically.

Reviewer 2 Report

The text would be more illustrative with the inclusion of some image that centered the pathological process in the anatomical territory. Also with some illustrative image of the surgical procedure

Author Response

Bovenkant formulier

Open Review

English language and style

( ) Extensive editing of English language and style required
( ) Moderate English changes required
( ) English language and style are fine/minor spell check required
(x) I don't feel qualified to judge about the English language and style

Yes

Can be improved

Must be improved

Not applicable

Does the introduction provide sufficient background and include all relevant references?

(x)

( )

( )

( )

Is the research design appropriate?

( )

(x)

( )

( )

Are the methods adequately described?

(x)

( )

( )

( )

Are the results clearly presented?

(x)

( )

( )

( )

Are the conclusions supported by the results?

(x)

( )

( )

( )

Comments and Suggestions for Authors

The text would be more illustrative with the inclusion of some image that centered the pathological process in the anatomical territory. Also with some illustrative image of the surgical procedure

Thank you for the positive review. I included two Figures, one explaining the anatomy of the different introital pain syndromes and another one showing the surgical technique. The latter was asked permission for re-use as it has been published in another (not-pubmed registered) journal.

Thanking you for the advice,

Warm regards

G Donders

Submission Date

12 December 2021

Date of this review

Bovenkant formulier

Open Review

English language and style

( ) Extensive editing of English language and style required
( ) Moderate English changes required
( ) English language and style are fine/minor spell check required
(x) I don't feel qualified to judge about the English language and style

Yes

Can be improved

Must be improved

Not applicable

Does the introduction provide sufficient background and include all relevant references?

(x)

( )

( )

( )

Is the research design appropriate?

( )

(x)

( )

( )

Are the methods adequately described?

(x)

( )

( )

( )

Are the results clearly presented?

(x)

( )

( )

( )

Are the conclusions supported by the results?

(x)

( )

( )

( )

Comments and Suggestions for Authors

The text would be more illustrative with the inclusion of some image that centered the pathological process in the anatomical territory. Also with some illustrative image of the surgical procedure

Thank you for the positive review. I included two Figures, one explaining the anatomy of the different introital pain syndromes and another one showing the surgical technique. The latter was asked permission for re-use as it has been published in another (not-pubmed registered) journal.

Thanking you for the advice,

Warm regards

G Donders

Submission Date

12 December 2021

Date of this review

29 Jan 2022 10:02:04

Onderkant formulier

© 1996-2022 MDPI (Basel, Switzerland) 

29 Jan 2022 10:02:04

Onderkant formulier

© 1996-2022 MDPI (Basel, Switzerland) 

Round 2

Reviewer 1 Report

The current version is very clear and informative and the introduction provides a thorough description of this entity.

As the authors discuss in the manuscript, the common diagnosis of vestibulodynia ignores, too often, diagnosable and treatable causes of introital pain.

This paper adds to previous reports which discuss introital dyspareunia (see Goldstein, "Female sexual Pain Disorders, Chapter 19, "Diagnostic and Treatment Algorithm for Women with Vulvodynia and Sexual Pain Disorders"  as well as the paper by Lev-Sagie et al in JCM, https://www.mdpi.com/2077-0383/9/7/2023, which discusses the allocation of vestibular pain into subgroups according to localization of pain in the vestibule, i.e., posterior-only (5 and 7 o'clock) versus circumferential (2-10-5 and 7 o'clock) provoked pain- and is in line with your description (Figure 1). Together, these reports challenge the current concept of "unidentified pathology" of introital pain.

This study, albeit its retrospective design, is adding practical knowledge and hopefully will assist physicians treating women with dyspareunia in providing better care. I think this is an excellent example of how an experienced clinician notices subgroups of patients and provides a different approach to diagnose and treat a very common problem, which is often misdiagnosed, mistreated, and mispresented in the current literature.